# Assessment of fire emissions inventories during the South American Biomass Burning Analysis (SAMBBA) experiment

G. Pereira[1], R. Siqueira[2], N. E. Rosário[3], K. L. Longo[2], S. R. Freitas[2], F. S. Cardozo[1], J. W. Kaiser[4], M. J. Wooster[5,6]

[1]Department of Geoscience, Federal University of Sao Joao del-Rei (UFSJ), Sao Joao del-Rei, Brazil
[2]Center for Weather Forecast and Climate Studies, National Institute for Space Research (INPE), Cachoeira Paulista, Brazil
[3]Environmental Sciences Department, São Paulo Federal University (UNIFESP), Diadema, São Paulo, Brazil
[4]Max Planck Institute for Chemistry (MPI-C), Mainz, Germany
[5]King's College London (KCL), London, United Kingdom
[6]NERC National Centre for Earth Observation (NCEO), United Kingdom

*Correspondence to*: G. Pereira (pereira@ufsj.edu.br)

**Abstract.** Fires associated with land use and land cover changes release into the atmosphere large amounts of aerosols and trace gases. Although there are several inventories of biomass burning emissions covering Brazil, there are still considerable uncertainties and differences among these. While most fire emissions inventories still utilize the parameters of vegetation fuel load, emission factors and other parameters to estimate the biomass burned and its associated emissions, certain more recent inventories tend to apply an alternative method based on fire radiative power (FRP) observations to estimate the amount of biomass burned and the corresponding emissions of trace gases and aerosols. The Brazilian Biomass Burning Emission Model (3BEM) and Fire Inventory from NCAR (FINN) are examples of the first, while Brazilian Biomass Burning Emission Model with FRP assimilation (3BEM_FRP) and Global Fire Assimilation System (GFAS) are examples of the latter method mentioned. In this paper, the output of four biomass burning emission inventories used during the South American Biomass Burning Analysis (SAMBBA) field campaign are analyzed and intercompared, focusing on eight pre-defined grids. Aerosol optical thickness ($AOT550_{nm}$) derived from measurements made by the Moderate Resolution Imaging Spectroradiometer (MODIS) sensor operating onboard Terra and Aqua satellites is applied to assess the model simulations consistency. Concerning the daily areal-averaged emission load of Carbon Monoxide (CO) to South America (01 Sep 2012 – 31 Oct 2012), a significant linear correlation ($r$, $p > 0.05$ level, Student t-test) between 3BEM and FINN, and between 3BEM_FRP and GFAS, were found with approximately 0.86 and 0.85, respectively. These results indicate that emissions estimates in this region derived via similar methods tend to agree with one other, but differ more from the estimates derived via the alternative approach. However, the evaluation of MODIS $AOT550_{nm}$ with linear correlations of the daily areal-averaged emission load of CO to eight specific grids indicate that 3BEM and FINN typically underestimated the smoke emission loading in the eastern region of Amazon Forest, whilst 3BEM_FRP presents a tendency to overestimate fire emissions in the same area. The consistence of the daily areal-averaged emission load of CO to South America to 3BEM and FINN fire inventories present a linear correlation coefficient of 0.75-0.92, with a tendency of FINN to overestimate the emission of carbon monoxide by 20-30%. Moreover, measure of accuracy of the daily areal-averaged emission load of CO to South America to 3BEM_FRP and

GFAS shows a linear correlation coefficient of between 0.82-0.90, with higher values near the arc of deforestation in Amazon Rainforest. However, GFAS has a tendency to present higher carbon monoxide emissions in this region, and 3BEM_FRP tends to overestimate the emissions in soybean expansion (east of Amazon forest). Furthermore, the evaluation of MODIS AOT550nm Terra and Aqua with the simulated $AOT550_{nm}$ from Monitoring Atmospheric Composition and Climate (MACC)

from European Centre for Medium-Range Weather Forecasts (ECMWF) and Coupled Aerosol and Tracer Transport model to the Brazilian developments on the Regional Atmospheric Modelling System (CCATT-BRAMS) operational atmospheric chemistry transport models presented a good agreement related with general pattern of the $AOT550_{nm}$ time-series. However, when the fires present a high biomass consumption, the aerosol values simulated by the two models underestimates the MODIS measurements.

**1 Introduction**

Biomass burning is a global phenomenon, and an ancient practice of human occupation, as well as a natural process. It consumes large amounts of vegetation across wide areas and modifies Earth surface characteristics. Fires still nowadays play a key role in ecosystem services, opening areas for livestock and agriculture, and pest control (Shimabukuro et al., 2013). In the last five decades biomass burning has been extensively and persistently used all over the tropics for these purposes, and

has been involved in widespread deforestation and forest degradation (Crutzen, 1990; Bustamante et al., 2015). Biomass burning emissions inject a considerable amount of oxidants and aerosols into the atmosphere, modifying atmospheric composition and reactivity therefore disturbing the regional climate, water and biogeochemical cycles (Andreae et al., 2002; Bowman et al., 2009). Moreover, fire emissions in tropical areas are felt not only regionally but rapidly and strongly affects the global scale due to the efficiency of the atmospheric transport processes of trace gases and aerosols emitted along with

high heat release, reinforced by the intense tropical convective activity associated with high temperature and moist (Kaufman et al., 1995; Martin et al., 2010). Extensive fires activity also disturbs the environmental system, producing soil depletion, damaging flora and fauna, decreasing biodiversity, and even affecting human life (Fearnside, 2000).

An accurate temporal and spatial estimate of biomass burning emissions is critical to a reliable analysis of the associated effects, including at different time scales. Nowadays, efforts to quantify emissions from biomass burning from space-borne

instruments have increased considerably in scope, but uncertainties on the input data and within the different methodologies can still lead to lead to errors of up to an order of magnitude for trace gases and aerosol emission totals (Vermote et al., 2009; Baldassarre et al., 2015).

In literature, is possible to find several products of burned area and fire emissions inventories, such as the Global Fire Emissions Database (GFED, van der Werf et al., 2010), the Brazilian Biomass Burning Emission Model (3BEM, Longo et al., 2010), the

Global Fire Assimilation System (GFAS, Kaiser et al., 2009/2012) and the Fire INventory from NCAR (FINN, Wiedinmyer et al., 2011). Most of the fire emissions inventories utilizes active fire locations and burned area to estimate the trace gases and aerosol emissions released into the atmosphere (Mao et al., 2014). However, the temporal and spatial distribution of biomass

burning emissions present several sources of errors related to the lack of detection of small fires, for example, prescribed and agricultural burning. Also, global burned area products are unsuitable to estimate the burned area of small fires due to limitations on its algorithms (Giglio et al., 2006; Giglio et al., 2010; Randerson et al., 2012).

Thus, most methodologies to estimate biomass burning emissions utilize the relationship proposed by Seiler and Crutzen (1980), usually called the "bottom-up approach":

$$M^{[\in]} = A.B.\beta.EF^{[\in]} \tag{1}$$

where $M^{[\in]}$ is the emission load of species $\in$ (g); A is the burned area (in km²); B is the fuel load (kg.km$^{-2}$); $\beta$ is the combustion completeness (unitless); and $EF^{[\in]}$ is the emission factor released species $\in$ (g.kg$^{-1}$). In this method, the burned area is one of the parameters necessary to estimate the emission of trace gases and aerosols, usually estimated by Earth Observation (EO) satellites, and typically since fires must have already occurred to produce a burned area this is not that well suited to providing near real-time air quality forecasts in models that includes vegetation fires. Other factors in Equation 1 can also be difficult to determine, for example, the combustion completeness typically depends on the moisture present in the biomass (and thus in part on micrometeorology), and the fuel load (average biomass and its fraction above ground) is hardly homogeneous (Chuvieco et al., 2004; Yebra et al, 2008; De Santis et al., 2010).

In the last decade, the development of sensors more targeted at active fire observations has allowed estimation of the radiative energy flux released by fires or the fire radiative power (FRP, in J.s$^{-1}$). The FRP accuracy has been evaluated by Kaufman et al. (1998) and Wooster et al. (2003) showing an average error of 16% and 6.5%, respectively. However, the error could be larger due the spatial resolution of several sensors, basically, the atmospheric transmittance and the cloud obscuration can cause an omission error of 15% and 11% in FRP estimates, respectively (Schroeder et al., 2008). Also, according to Vermote et al. (2009), the integration of FRP over the fire cycle and its conversion to burned biomass can cause an error of 21% and 10%-30%, respectively, varying according to heterogeneity of regional/zonal characteristics.

Continuous acquisition of FRP over a fires duration provides fire radiative energy (FRE, in MJ) released by the fire process. New fire emission methods utilize the FRE to calculate the amount of biomass burned and/or the emission of trace gases and aerosols (Wooster et al., 2005; Ichoku and Kaufman, 2005; Ellicot et al., 2009; Freeborn et al., 2011; Kumar et al., 2011):

$$FRE_{grid_{(lon,lat)}} = \frac{1}{2}\sum_{i=1}^{n}(FRP_n + FRP_{n+1}).(T_{n+1} - T_n) \tag{2}$$

$$M^{[\in]} = FRE_{grid_{(lon,lat)}}.\gamma.EF^{[\in]} \tag{3}$$

where $FRE_{grid_{(lon,lat)}}$ provides radiative energy at a geographical location in terms of longitude and latitude of a specific centered point of a regular grid; T is the time interval between FRP acquisitions; n represents the n[th] sample; and $\gamma$ is the FRP-biomass conversion factor (kg.MJ$^{-1}$, Wooster et al., 2005; Kaiser et al., 2012). In this method, parameters such as fuel load, burning efficiency and the presence of moisture in the soil and in vegetation directly influence the observed energy radiated by the fires and do not have to be separately considered. In this way this method is sometimes referred to as a "top-down" approach.

The South American Biomass Burning Analysis (SAMBBA) was an airborne experiment design to characterize the smoke physical and chemical properties in Amazonian rainforest and central area of Brazil. The SAMBBA campaign took place in September 2012. The operational smoke forecasting system built to support SAMBBA flights planning utilized four fire emissions inventories that deployed variously the aforementioned bottom up and top down approaches: I) Brazilian Biomass Burning Emission Model (3BEM, Longo et al., 2010); II) Brazilian Biomass Burning Emission Model with FRP assimilation (3BEM_FRP, Pereira et al., 2009); III) Global Fire Assimilation System (GFAS, Kaiser et al., 2012); and IV) Fire Inventory from NCAR (FINN, Wiedinmyer et al., 2011). This study provides an intercomparison and evaluation of these inventories, with focus on the South American Biomass Burning Analysis (SAMBBA) field campaign experiment.

## 2 Data and methodology

### 2.1 Inventories description

#### 2.1.1 Brazilian Biomass Burning Emission Models (3BEM and 3BEM_FRP)

The 3BEM is a model developed to estimate the daily fire emissions based on the location of actively burning fire "hotspots" (i.e. areas of combustion detected using active fire/thermal anomaly detection algorithms) derived via orbital remote sensing. The 3BEM utilizes the Wildfire Automated Biomass Burning Algorithm (WFABBA) applied in Geostationary Operational Environmental Satellite (GOES) constellation data (Prins et al., 1998); the MOD14 and MYD14 products of the Moderate Resolution Imaging Spectroradiometer (MODIS) aboard Terra and Aqua satellites (Justice et al., 2002); and the fire product developed by the Environmental Satellite Division (DSA) of National Institute for Space Research (INPE), which uses the Spinning Enhanced Visible and Infrared Imager (SEVIRI) onboard Meteosat Second Generation (MSG), GOES, MODIS and Advanced Very High Resolution Radiometer (AVHRR) onboard the National Oceanic and Atmospheric Administration (NOAA) constellation (Setzer and Pereira, 1991).

The 3BEM version initially developed by Longo et al. (2010), estimates the emitted mass of trace gases and aerosols related to each fire detected by remote sensing as described in Equation 1. Therefore, the location of the detected fire hotspots are cross-tabulated with MODIS Land Cover map to allow the estimation of above-ground biomass density, combustion factor and emission factor from literature values (Olson et al., 2000; Andreae and Merlet, 2001; Houghton et al., 2001; Sestini et al., 2003; Akagi et al., 2011). Also, the model filters all fires located in a 1 km radius to prevent double counting between fire products.

The updated 3BEM includes FRP assimilation (3BEM_FRP, Pereira et al., 2009), and utilizes the same algorithm of the former 3BEM but with burned biomass directly estimated using FRE estimates, as described in Equation 3. The 3BEM_FRP model groups all FRP values estimated by MODIS, GOES and METEOSAT products according to their time acquisition, eliminating the low confidence fire pixels (values below 50% for MOD14, MYD14 and METEOSAT products, and flags 4 and 5 for WFABBA/GOES product) and minimizing the impact of the MODIS bow-tie effect as described in Freeborn et al. (2011).

Also, due to the frequency of observations (mainly in GOES and SEVIRI data) 3BEM needs only one fire detection to estimate the biomass burned and its associated emissions due to the filtering process (Longo et al., 2010). In 3BEM_FRP if the active fire has no subsequent observations in next four hours the algorithm assume that the fire event is over. Thus, missed detection due cloud have impact in FRE integration if the cloud persists for more than 8 satellite acquisitions.

Some of these FRP estimates are compromised by sensor saturation over larger fires, particularly GOES over South America (Xu et al., 2010). Thus, to not ignoring important episodes of biomass burning by removing GOES saturated pixels, which FRP values are not provided, 3BEM_FRP utilizes Eq. (4) to estimate the energy released by fires, based on the premise that emitted spectral radiance ($M_\lambda$) in spectral band centered at 3.9 µm is proportional to FRP (Wooster et al., 2003).

$$FRP_{MIR} = \frac{Ag}{a} \sigma \int_{3.76}^{4.03} M(\lambda, T) d\lambda - M_b \tag{4}$$

where $Ag$ represents the area of GOES pixel (km²); $a$ is a constant fit based in GOES MIR spectral band (W.m$^{-2}$.sr$^{-1}$.µm$^{-1}$.K$^{-4}$, Wooster et al., 2005); $\sigma$ is the Stefan-Boltzmann constant (5.66 x 10$^{-8}$ W.m$^{-2}$.K$^{-4}$); $M$ is the Planck's Law (W.m$^{-2}$.µm$^{-1}$); $\lambda$ is the wavelength (µm); T represents the temperature (K); and $M_b$ is the radiance emitted by the background (110 MW). Also, in 3BEM_FRP methodology, FRP values estimated by GOES satellites below 1000 MW are corrected by +17% and FRP values higher than 1000 MW are corrected by +41% (Xu et al., 2010). This procedure is also applied in SEVIRI data, but due to its

spatial coverage, we decided not to include this data in the present analysis. Pereira et al. (2009) describes the 3BEM_FRP method in detail.

A clustering process performs the combination of all detected fires from different sensors. In this step, the size of a matrix that merges FRP data can be defined according to the spatial resolution and grid configuration of the atmospheric model. Consequently, the convolution mask $\eta(\gamma,\kappa)$, of size M x N (rows x columns), running over the grid with FRP areal density

(defined by weighting the FRP values by pixel area) values estimated by different satellites $\xi(lon,lat)$, will result in the grid ($FRP_{grid}$) containing all clustered fires for a given timestep.

$$FRP_{grid(lon,lat,t)} = \sum_{\gamma=-\alpha}^{\alpha} \sum_{\kappa=-\beta}^{\beta} \eta(\gamma,\kappa) \xi(lon + \gamma, lat + \kappa, t) \tag{5}$$

where the clustered grid is defined to all points where the mask of M x N size overlaps the image completely (lon $\epsilon$ [α, M −α], lat $\epsilon$ [β, N −β]). Moreover, if in any timestep of FRP integration the interval between two acquisitions is greater than 4 hours

($\Delta$T > 14400 s), it is assumed the hypothesis of two or more independent fires, then the algorithm assumes T = 0 and another integration will be performed.

### 2.1.2 Global Fire Assimilation System (GFAS)

The GFAS is an approach used to map daily global fire emission through FRP observations. Therefore, GFAS also assumes that the electromagnetic radiation emitted by fires is related to the consumption of burned biomass (Wooster et al., 2005). In

the present GFAS version, 1.1, FRP values of MOD14 and MYD14 fire products from Terra and Aqua satellites, respectively, are used to estimate the average of observed FRP areal density ($FRP_{ad}$, in W.m$^{-2}$). GFAS estimates open vegetation fire trace gas and particulate emissions from each fire detected as described in Eq. (3).

The model performs a clustering process of observed FRP ($F_i$), pixel area ($A_i$) and view zenith angle ($\theta$) to sensor pixels with valid observations (i) to estimate the $FRP_{ad}$. The clustering process also takes observations of $F_i=0$, i.e. no-fire, into account. Thus, for each individual grid cell the estimated $FRP_{ad}$ could be calculated as:

$$FRP_{ad} = \frac{\sum_k \sum_{i_k \epsilon j} F_{i_k.} \cos^2(\theta_{i_k})}{\sum_k \sum_{i_k \epsilon j} A_{i_k.} \cos^2(\theta_{i_k})}$$

where $i_k$ represents the pixel $i$ of satellite product $k$ (MOD14, MYD14). This formulation implicitly corrects the MODIS bow-tie effect and partial cloud/ice/snow/water-cover of a grid cell. Observation gaps are subsequently filled with a data assimilation approach, that currently assumes persistence of the fires.

In GFAS, the coefficient that converts the $FRP_{ad}$ to dry matter combustion rate is based on eight land cover classes, cf. Heil et al. (2010); Kaiser et al. (2012). In addition, the emission load of 40 species is calculated using emission factors from Andreae
and Merlet (2001), subsequent updates and Christian et al. (2003). Kaiser et al. (2012) describes the inventories method in detail.

### 2.1.3 Fire INventory from NCAR (FINN)

The FINN is an approach to estimate daily fire emissions at 1 km resolution through satellite observation of active fires. The FINN model produces global estimates of aerosols and trace gases of open vegetation fires, as described in Equation 1. The
FINN utilizes the MOD14 and MYD14 fire products, processed by MODIS Rapid Response (MRR) or by the MODIS Data Processing System (MODAPS) Collection 5. However, since MODIS observations do not cover the entire globe daily, due to orbital gaps, FINN smears MODIS detections of active fire over two days. For each fire located in the equatorial region, is assumed that in the next day the fire will be half of its original size (Wiedinmyer et al., 2011).

Similarly to 3BEM, the FINN model removes multiple detections of same fire pixel prior to estimation of the trace gases and
aerosols released. In addition, for each active fire, FINN estimates as 1 km² the burned area for most of land use classes, with exception to grasslands/savannas in which it is assumed the burned area as 0.75 km². To estimate the emission of trace gas and aerosol species, emissions factors derived from Andreae and Merlet (2001) and Akagi et al. (2011) are used, as well as the MODIS land cover type for 2005 and the MODIS Vegetation Continuous Fields (VCF) product. Wiedinmyer et al. (2011) describes the procedures adopted in detail.

**2.2 MODIS aerosol optical thickness**

The aerosol optical depth (AOD) product from MODIS sensors aboard Aqua and Terra satellites platforms is used to provide a first order assessment of the impacts of the two distinct methods to estimate biomass burning emission, i.e. the bottom-up and top-down approaches. In this work, the MODIS Level 2.0 Collection 5.1 (051) data and Level 3 Atmospheric product denominated MYD08_D3 (Mean Aerosol Optical Thickness at 550 nm – $AOT_{550nm}$) is utilized to compare the fire emissions
inventories utilized during the SAMBBA campaign between 01 Sept and 31 Oct 2012. The MODIS Atmosphere daily global product estimates roughly 600 statistical datasets at 1 degree resolution and equal-angle grid that comprehends a merge of

MODIS acquisition over the globe (Kaufman and Tanré, 1998). Figure 1a shows the MODIS MCD12 Land Use and Land Cover (LULC) product for South America with LULC classes described in Table 1. During the SAMBBA field campaign, the highest biomass burning aerosol loadings were observed over Brazilian territory, mainly over the southeastern edge of the Amazonian forest (dotted red line, known as Arc of Deforestation) and in the soybean expansion area in the Brazilian Savannah

(marked as X), as shown in the time-averaged (01 Sep 2012 – 31 Oct 2012) $AOD_{550nm}$ field derived from MODIS sensor aboard Aqua satellite (Figure 1b). The high values of AOD related to fires located in eastern of Mato Grosso (mainly in the secondary forest) and in transition areas of Amazon rainforest and Brazilian Savannah are noteworthy; soybean expansion areas, which present high concentration of fires, have a lower amount of biomass than Amazon rainforest and the high ventilation favors the transport of smoke to west.

**2.3 Inventories configuration and analysis description**

To evaluate the fire emissions inventories utilized during the SAMBBA experiment, we used the 3BEM preprocessor to generate the gridded data in Geographical coordinates with a spatial resolution of 0.1°. The 3BEM preprocessor has as output daily emission load of several species, but in this work, we selected to analyze only carbon monoxide (CO) emission fields, from 1st of September to 31st of October 2012. Moreover, the four inventories are compared over eight sub-domains windows

with a mark of 10°x10° that typically comprehends different Brazilian states (Figure 2a) and different fire regimes and biomes (Figure 2b).

Two chemistry transport models were selected to applied in real time during the SAMBBA experiment, the Monitoring Atmospheric Composition and Climate (MACC) from European Centre for Medium-Range Weather Forecasts (ECMWF) with GFAS fire emission inventory (described in section 2.1.2) and the Coupled Aerosol and Tracer Transport model to the

Brazilian developments on the Regional Atmospheric Modelling System (CCATT-BRAMS) with 3BEM fire emission inventory (described in section 2.1.1). These models were selected due to distinct methodology to estimate the biomass burning emissions (the first a

top-down approach, and the second the bottom-up approach). The MACC/ECMWF were applied with a global domain, 1.0° of horizontal resolution, 20 levels of vertical resolution, assimilation operation mode, forecast frequency of 06 hours, and

boundary conditions from ECMWF model. The CCATT-BRAMS were applied with South America domain, 0.22° of horizontal resolution, 24 levels of vertical resolution, forecast operation mode, forecast frequency of 03 hours, and boundary conditions from Center for Weather Forecasting and Climate Research (CPTEC) model.

**3 Results and discussion**

**3.1 Comparing emission inventories spatially**

During the SAMBBA experiment, four inventories were used in near real time within atmospheric-chemistry transport models to support the SAMBBA flight planning (FINN, GFAS, 3BEM, 3BEM_FRP). Each inventory's spatial distribution of total

CO emission ($10^4$ kg.m$^{-2}$) over South America, from $1^{st}$ of September to $31^{st}$ of October 2012, is depicted in Figure 3, according to the respective methodologies used to estimate the emission loading.

The intercomparison shown in Figure 3 demonstrates that inventories that utilize the same (i.e. top-down or bottom-up) methodology show similar spatial patterns in CO emissions (kg.m$^{-2}$), not only in Amazon basin but across all of South America, but differ in their absolute values. The CO emissions estimated by the 3BEM and FINN emission inventories (Figure 3a and 3b, respectively) present higher values than 3BEM_FRP (Figure 3c) and GFAS (Figure 3d) in the regions where the main processes of forest logging and subsequent agricultural expansion (GRIDS 3, 4 and 5) occur. The highest emissions in 3BEM and FINN are located mainly in Rondonia State (GRID3) and in the border of Mato Grosso State (GRID 4), where most of the SAMBBA flights took place.

Cardozo et al. (2014) analyzed the fires pattern in Rondonia between 2000 and 2011 and identified that most fires result in relatively small "burn scars" on the landscape, with areas 20 to 80 hectares (64% of cases). In addition, only 6.5% of all burned areas in Rondonia are associated with recently deforested areas. This could indicate that 3BEM and FINN are overestimating the CO emission load due to an erroneous location of fires in areas of forest instead of livestock and permanent crops, which have lower above ground biomass. Furthermore, 3BEM_FRP and GFAS inventories do not display a similar pattern to 3BEM and FINN because their emissions are directly related to FRP, with a weaker dependency on land cover type. Thus, areas with low above ground biomass will provide low values of FRE due the observed characteristics of the fire activity, and thus low values of CO.

The spatial distribution of the emission inventories suggests that fires in the region are strongly related to deforestation activity and therefore to the general economy, with a strong trend in recent years of fires in secondary forest (Cardozo et al., 2014). In Figure 3c, higher emission loads are located in the east of Brazil (GRID 5), mainly in the Cerrado biome, a vegetal formation composed of savanna and other typically low density vegetation formations, which include trees of 15 meters height (as shown in Figure 1a). This region is now economically used, constituting a new agricultural frontier of Brazil (with more than 100 million hectares suitable for modern mechanized crop agriculture, mainly soybean). In this area, the four fire inventories differ considerably. The 'top down' inventories that use the FRP approach show much higher emission loads compared to 3BEM and FINN.

The difference in South America daily areal-averaged emission of CO (kg.m$^{-2}$) between $1^{st}$ of September and $31^{st}$ of October 2012 was quantified via linear correlation coefficient analysis (all significant at $p < 0.05$) (Figure 4). The highest linear correlations coefficients were found between 3BEM and FINN with 0.86, between 3BEM_FRP and GFAS with 0.85, and between GFAS and FINN with 0.84. These high linear correlations indicate that inventories produced using similar emissions methods tend to agree with each other. The third correlation reflects that both inventories use the same active fire observations (MODIS), albeit with different data processing. We highlight that all were significant at $p>0.05$ level, Student t-test. To analyze the measures of accuracy of regression between fire inventories we used the bootstrap technique (Efron, 1982). In this technique, a population of $1.0 \times 10^4$ reconstructs the regression and provides the parameters to create the confidence interval and error analysis of model estimation.

The bootstrap regression among the daily areal-averaged CO values to 3BEM and FINN emission inventories present a linear correlation between 0.75-0.92 with a tendency of FINN to overestimate, relative to 3BEM, the emission load of CO in 20-30% (Figure 4). This apparent FINN overestimation is seen in the majority of grids, with the exception of grids 2, 5 and 7 (Figure 5a). In areas where FINN emissions are lower than 3BEM emissions, vegetation is mainly composed of Brazilian Cerrado (grids 2 and 5), and by Pantanal wetlands biome and croplands/livestock (grid 7) with predominant vegetation are Savannah and grassland, however, the presence of Deciduous and Semi deciduous Forest are also found. Since, 3BEM and FINN presents similar methodology to estimate the emission load of species released by wildfires, the parameters used in Equation 1, such as above ground biomass, are likely to be associated with the relatively high estimation in CO values.

The intercomparison analysis of the daily areal-averaged CO values to 3BEM and GFAS inventories shows a linear correlation coefficient between 0.75-0.85 with higher values over the Arc of Deforestation, in the Amazon Rainforest (Figure 1a). However, GFAS has a tendency to present higher CO emissions (by 10-20%) in grids 5 and 8, and to underestimate by 20-30% in grids 3 and 4 (Figure 5b) when compared to 3BEM. In these areas, the vegetation is dominated by the Amazon biome, along with a small area of Cerrado (located in the south of the grids), with presence of Evergreen Broadleaf forest, Tropical Degraded Forest, and Cropland/livestock areas. Moreover, the daily areal-averaged CO values to FRP-based emissions estimation method utilized by GFAS presents a considerable difference in the region of soybean expansion in grid 5.

Similar to GFAS, the relationship among the daily areal-averaged CO values to 3BEM and 3BEM_FRP shows a linear correlation coefficient between 0.65-0.75. However, 3BEM_FRP has a tendency to overestimate the CO values by 60-85% in grid 5, and to underestimate by 10-15% in grids 3 and 4 (Figure 5c). The 3BEM_FRP model presented an elevated emission of CO in grid 5, possibly due the GOES viewing zenith angle (high viewing angles results in the erroneous values of infrared brightness and present a large pixel area, Roberts et al., 2005; Vermote et al., 2009; Peterson et al., 2013). In addition, in grid 5, cloud absence may influence in the FRP cycle occasioning an overestimation due the large number of acquisitions in high view angles. The relationship between the daily areal-averaged CO values to 3BEM_FRP and GFAS shows a linear correlation between 0.82-0.90. Accordingly, the 3BEM_FRP model has a tendency to present higher emission load of CO than GFAS, mainly in some areas of deforestation arc and in grid 5 that could reach 100% (Figure 5d). The high values are associated with assimilation of GOES FRP in 3BEM_FRP, while GFAS utilizes only MODIS FRP data.

The fieldwork measurements acquired during the SAMBBA campaign indicate that the near-real time air smoke forecasts based on traditional inventories, such as 3BEM used by CCATT-BRAMS, typically underestimated smoke loading in the central to east region of Amazon Forest (near Mato Grosso and Tocantins States) as described in Rosario et al. (2013). In addition, the emission inventories tended to overestimate the smoke loading in the Northwest part of Rondonia (indicated in Figure 2a). However, during the SAMBBA campaign were identified lower values of CO emissions, showing an overestimation of emission load in this region. In general, all fire emission inventories present a good agreement, with most regressions significant at p>0.05 level, Student t-test (Table 2, non-significant regression marked in red). The only regression that presented low values of linear correlation coefficients are located in Grid 5 with values lower than 0.30. Moreover,

regression between 3BEMxFINN present the highest correlation in most of the grids. Also, the absolute bias analysis indicate a high variability in daily areal-averaged emission load of CO between the four fire inventories.

## 3.2 Assessment of Fire inventories with AOT

Figure 6 shows the daily emission estimates from each of the emission inventories used in SAMBBA campaign integrated over the eight grids and the areal-averaged values of $AOT_{550nm}$ for 01st September 2012 to 31st October 2012. During the SAMBBA campaign, the grids located in arc of deforestation and in Mato Grosso state presented the highest values of averaged $AOT_{550nm}$. Commonly, the 'top down' inventories that use FRP to estimate the emission of CO present a similar pattern, as do the 'bottom up' emission inventories that utilizes the relationship between the burned area, fuel load and the combustion completeness. This is evident in Figure 6a, which 3BEM_FRP and GFAS has lower values during all periods, with daily emission loads of CO between 20-500 $kg.m^{-2}$ and average value of 93 $kg.m^{-2}$. In addition, 3BEM and FINN emission inventories presented a high variability during the analyzed period, with the daily areal-averaged estimation of CO emission load that could reach values greater than 3000 $kg.m^{-2}$. The linear correlation coefficient of 3BEM_FRP x GFAS and 3BEM x FINN are 0.85 and 0.76, respectively, with a tendency of GFAS and 3BEM to present higher estimation of CO emission load. In the grid1, the total emission load of CO in 3BEM and FINN to 01st September 2012 to 31st October 2012 are, respectively, 61.660 and 52.971 $kg.m^{-2}$ (Table 3), an estimation 4-10 times more than GFAS and 3BEM_FRP.

Figure 6b shows the areal-averaged time series of CO and $AOT_{550nm}$ in grid2, with the four emission inventories presented showing similar patterns with high emission values on 09th September 2012 and between 20th-30th October 2012. The linear correlation coefficient of 3BEM_FRP x GFAS and 3BEM x FINN are 0.78 and 0.75 with a tendency of 3BEM (55.234 $kg.m^{-2}$) and 3BEM_FRP (29.059 $kg.m^{-2}$) to present higher estimation of CO emission load in this region. Also, in the general pattern of temporal evolution we could observe a good agreement between CO emission load estimated by the four emission inventories and MODIS $AOT_{550nm}$. However, the grid3 presented distinct CO distributions to the four emission inventories (Figure 6c). In September, FINN and 3BEM presented highest values when compared with 3BEM_FRP and GFAS during all period. However, 3BEM_FRP shows higher values than GFAS in September and otherwise in October. The linear correlation coefficient of 3BEM_FRPxGFAS and 3BEMxFINN are 0.87 and 0.94, respectively, with a tendency of FINN and 3BEM_FRP to present higher estimation of CO emission load. The relationship between GFASxFINN presents a linear correlation coefficient of 0.91. However, the total emission load of CO estimated by FINN (163.552 $kg.m^{-2}$) is 3 times more than GFAS (54.891 $kg.m^{-2}$).

In Grid4, all inventories showed a good agreement with linear correlations coefficients greater than 0.90. However, during the period from 3rd of September to 10th of September, 3BEM_FRP presented six episodes with higher emission loads, undetected in the other inventories (Figure 6d). The outlier values in 3BEM_FRP are likely to be related to GOES FRP acquisitions, which suggests inconsistencies in GOES data acquisition due to viewing angle and errors in data processing. The grid5 presented the lowest linear correlation coefficient between 3BEM and FINN emission inventories during the SAMBBA campaign with 0.30. In this grid, FINN presents a low estimation of CO emission load when compared with the other emission inventories (as

demonstrate in Table 3). In this region, FRP methods showed more emission than the bottom up methods. The 3BEM_FRP model overestimated in more than 100% the emission values compared to GFAS (Figure 6e), however, the linear correlation between these emission inventories is 0.84. In the other grids, the fire-related emission of CO provided by the four emission inventories show a considerable consistency, with few differences in absolute values, indicating a good agreement across these other regions of South America (Figures 6f-6h).

### 3.3 Evaluating the emissions against observations

During the SAMBBA campaign, operational atmospheric chemistry transport models were applied in real time: I) the MACC/ECMWF with GFAS; and II) CCATT-BRAMS with 3BEM. Figure 7 shows the simulated $AOT_{550nm}$ from MACC/ECMWF (red line), CCATT-BRAMS (blue line), and estimated by Terra (green line) and Aqua (black line) to the eight grids. Although 3BEM present the highest emission load in Grid1 and Grid2, the CCATT-BRAMS simulation underestimate the MODIS $AOT_{550nm}$. In these grids, the MACC/ECMWF model presents a better consistency with the satellite measurements. One potential reason for the underestimation of CCATT-BRAMS is related to background aerosol unrelated to the fires directly being observed and that is not included in this model, causing low values of simulated $AOT_{550nm}$ for these grids. Also, MACC/ECMWF assimilates MODIS aerosol optical depth observations, and even with low CO emission values in Grids 1 and 2 the aerosol simulation present a good agreement with observation data. Over the Grid5, the CCATT-BRAMS AOT underestimation is likely to be associated with biomass burning emission issues, since major contribution to AOT in that region come from smoke particles. 3BEM, which was used to feed CCATT-BRAMS, and FINN emission inventories, both bottom up based, presented much lower emission in Grid5 than FRP based inventories.

The assessment of MACC/ECMWF and CCATT-BRAMS models against MODIS $AOT_{550mm}$ reveals a good agreement in terms of the general pattern of the temporal evolution. As shown in the $AOT_{550mm}$ time series, both methods appear to be able to estimate the influence of aerosols released by fires rather well. However, when the intensity of the biomass burning is too high, the values simulated by MACC/ECMWF and CCATT-BRAMS for the grids 4-8 appear underestimated, possibly due to the influence of smoke on the FRP measurements, lack of fire observations, clouds, above ground biomass data and fire size. Moreover, it is possibly to identify an overestimation in Grid3, mainly due to out of date land use and land cover map that insert fires over forest areas even if the area was burned/deforested in past years, and a very noticeable underestimation in Grid5, as demonstrated in the inventories.

### 4 Conclusions

Full characterization of the emissions of trace gases and aerosols are often essential for assessing the atmospheric impacts of fire and for constructing fire inventories. These inventories generally rely on data from environmental satellites, and provide useful information for weather, climate and air quality models. In this study, we analyzed data from the four biomass burning emissions inventories used during the SAMBBA airborne atmospheric sampling and remote sensing campaign that took place

in Rondonia (Brazil) between 1 September 2012 and 31 October 2012. Each inventory utilizes distinct methodology, with 3BEM and FINN deriving the emission of trace gases and aerosols through a 'bottom up' combination of burned area and fuel load metrics based on vegetation maps and field-location specific coefficients, whilst 3BEM_FRP and GFAS estimate biomass burned more directly from 'top down' FRP measurements made by the EO satellite instruments.

The evaluation of the emission inventories focused on eight pre-defined grids, (Figure 2) and demonstrates that inventories that utilize the same methodology, such as 3BEM and FINN on the one hand ('bottom up') and GFAS and 3BEM_FRP on the other ('top down'), show similar patterns in emission spatial distribution, not only in the deforestation arc but also throughout South America. However, they can differ in their absolute values. As such, each inventory particular characteristics, with 3BEM and FINN showed higher emissions of CO values in the Amazon forest logging area, where most of the SAMBBA

campaign flights occurred. Furthermore, these emission inventories typically underestimated the smoke loading in the southeast region of Amazon Forest and in the northwest of Rondonia, where 3BEM_FRP and GFAS present higher values of emission load of CO. The best overall linear correlations coefficients were found between 3BEM and FINN, with approximately 0.86 and 3BEM_FRP and GFAS, with approximately 0.85, which indicate that similar emissions methods tend to agree with each other. Furthermore, the evaluation between the 3BEM and FINN fire inventories present a linear correlation

coefficient between 0.75-0.92, with total emission of CO estimated by FINN greater than 3BEM in grids 3,4,6 and 8. Moreover, 3BEM_FRP and GFAS shows a linear correlation coefficient between 0.65-0.95, with total emission of CO estimated by GFAS greater than 3BEM_FRP in grids 1, 6 and 8.

During the SAMBBA campaign, the assessment of simulated $AOT_{550nm}$ from MACC/ECMWF and CCATT-BRAMS operational atmospheric chemistry transport models with MODIS $AOT_{550nm}$ Terra and Aqua measurements show a good

agreement related with general pattern of the time-series. Also, MACC/ECMWF and CCATT-BRAMS models are capable to simulate the aerosols released by fires. However, when the intensity of the biomass burning is too high, the aerosol values simulated by the two models underestimates the MODIS measurements.

## 5 Acknowledgements

We would like to thank the São Paulo Research Foundation (FAPESP) for their financial support (2012/13575-9) and Minas

Gerais State Research Foundation (FAPEMIG, grant number: APQ-01698-14). J.W.K. and M.J.W were supported by NERC in the SAMBBA project (Grant number: NE/J010073/1).

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

Table 1. LULC classes based on Global Land Cover Characterization (GLCC).

| 1 | Crop / Mixed farming | 11 | Semi-desert |
|---|---|---|---|
| 2 | Short grassland | 12 | Ice cap/glacier |
| 3 | Evergreen needleleaf tree | 13 | Bog or marsh |
| 4 | Deciduous needleleaf tree | 14 | Inland water |
| 5 | Deciduous broadleaf tree | 15 | Ocean |
| 6 | Evergreen broadleaf tree | 16 | Evergreen shrub |
| 7 | Tall grass | 17 | Deciduous shrub |
| 8 | Desert | 18 | Mixed Woodland |
| 9 | Tundra | 19 | Forest/Field mosaic |
| 10 | Irrigated Crop | 20 | Water and Land mixture |

Table 2. Linear Correlation coefficients (r) between 3BEM, 3BEM_FRP, FINN and GFAS daily areal-averaged emission inventories to the eight grids and average absolute bias of CO (in kg.m$^{-2}$). In the table, the first value indicates the r and the second represents the absolute bias (r/bias).

| | 3BEM_FRP | FINN | GFAS |
|---|---|---|---|
| 3BEM_G1 | 0.84 / 933 | 0.85 / 145 | 0.82 / 736 |
| 3BEM_FRP_G1 | | 0.67 / -788 | 0.76 / -197 |
| FINN_G1 | | | 0.82 / 591 |
| 3BEM_G2 | 0.70 / 436 | 0.75 / 458 | 0.79 / 210 |
| 3BEM_FRP_G2 | | 0.61 / 22 | 0.78 / -226 |
| FINN_G2 | | | 0.82 / -249 |
| 3BEM_G3 | 0.78 / 1642 | 0.94 / 1674 | 0.90 / -136 |
| 3BEM_FRP_G3 | | 0.77 / 32 | 0.87 / -1778 |
| FINN_G3 | | | 0.91 / -1811 |
| 3BEM_G4 | 0.81 / -245 | 0.93 / 996 | 0.90 / -263 |
| 3BEM_FRP_G4 | | 0.71 / 1241 | 0.90 / -18 |
| FINN_G4 | | | 0.88 / -1260 |
| 3BEM_G5 | **0.16** / -6345 | 0.30 / -793 | 0.36 / 605 |
| 3BEM_FRP_G5 | | 0.47 / 5552 | 0.84 / 6950 |

| | | | |
|---|---|---|---|
| **FINN_G5** | | | 0.80 / 1399 |
| **3BEM_G6** | 0.46 / 12 | 0.89 / 68 | 0.88 / -86 |
| **3BEM_FRP_G6** | | 0.32 / 55 | 0.65 / -99 |
| **FINN_G6** | | | 0.81 / -154 |
| **3BEM_G7** | 0.68 / -14 | 0.89 / 151 | 0.85 / 413 |
| **3BEM_FRP_G7** | | 0.60 / 165 | 0.79 / 427 |
| **FINN_G7** | | | 0.85 / 261 |
| **3BEM_G8** | 0.63 / -138 | 0.84 / -251 | 0.79 / -31 |
| **3BEM_FRP_G8** | | 0.65 / -113 | 0.65 / 106 |
| **FINN_G8** | | | 0.91 / 219 |

Table 3. Total mean emissions (kg.m$^{-2}$) of CO in the eight grids shown in Figure 2.

| | Total Emission of CO (kg.m$^{-2}$) | | | |
|---|---|---|---|---|
| | **3BEM** | **FINN** | **GFAS** | **3BEM_FRP** |
| **GRID1** | 61.660 | 52.971 | 17.485 | 5.636 |
| **GRID2** | 55.234 | 42.631 | 27.706 | 29.059 |
| **GRID3** | 155.346 | 163.552 | 54.891 | 56.815 |
| **GRID4** | 151.720 | 167.518 | 91.905 | 166.423 |
| **GRID5** | 82.271 | 45.945 | 129.866 | 274.896 |
| **GRID6** | 35.199 | 40.412 | 31.118 | 29.811 |
| **GRID7** | 53.072 | 28.266 | 43.981 | 48.541 |
| **GRID8** | 11.342 | 13.252 | 25.674 | 19.633 |
| **TOTAL** | 605.844 | 554.547 | 422.626 | 630.814 |

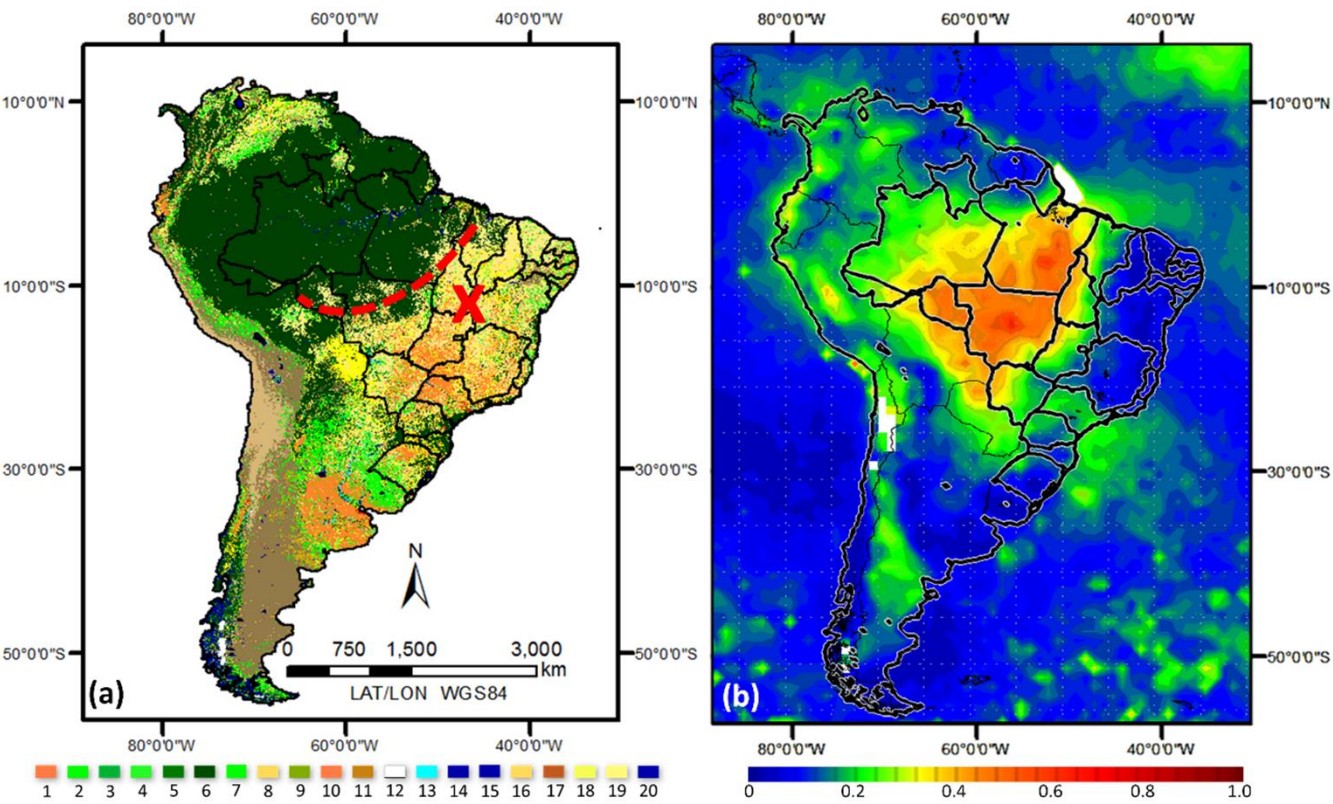

Figure 1. (a) MCD12 Land Use and Land Cover (LULC) for South America with dotted red line representing the Amazonian Arc of Deforestation and soybean expansion in Brazilian Cerrado (marked as X); (b) Time-averaged Aqua MODIS aerosol optical thickness at 550 nm during 01 Sep 2012 – 31 Oct 2012 over South America.

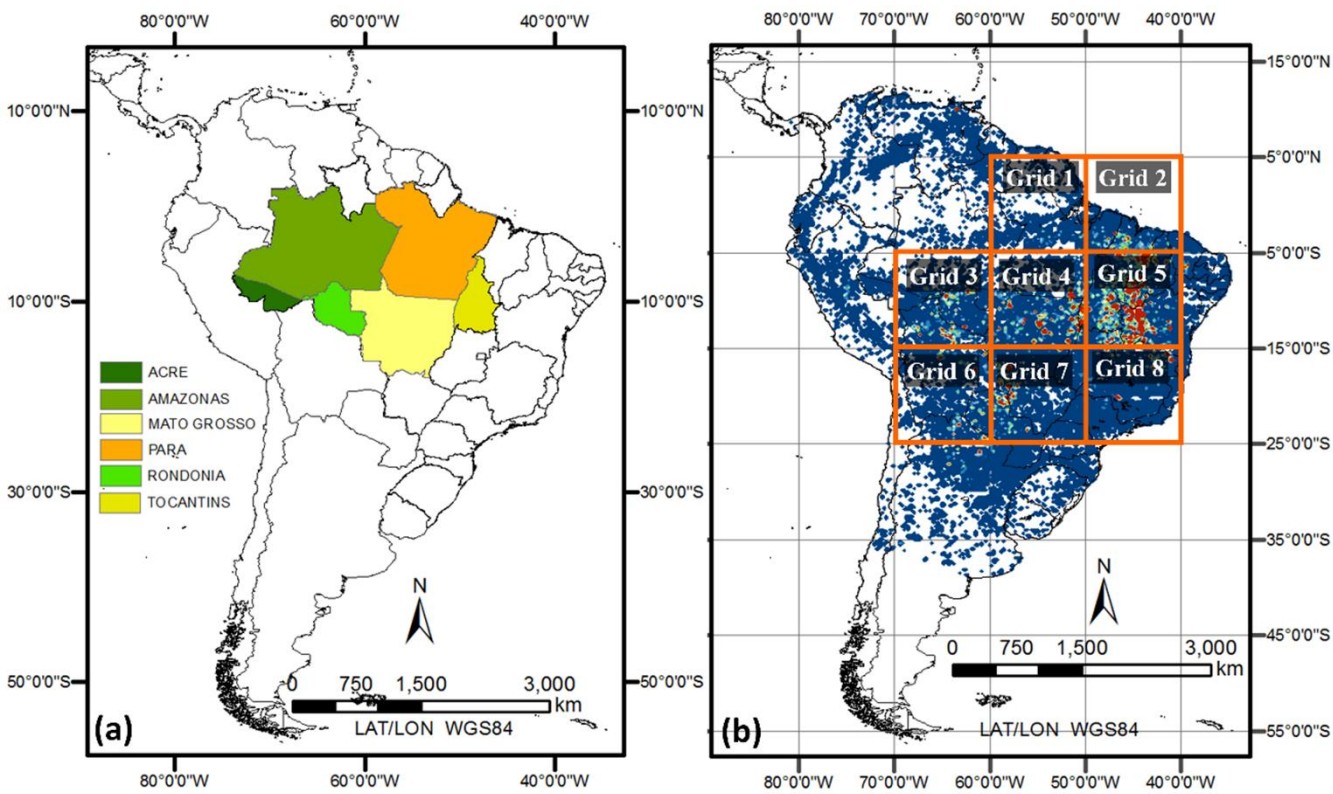

Figure 2. South America maps highlighting (a) the Brazilian states of Acre, Amazonas, Mato Grosso, Para, Rondonia and Tocantins, where SAMBBA field campaign took place; and (b) the eight sub-domains windows for the biomass burning inventories comparison.

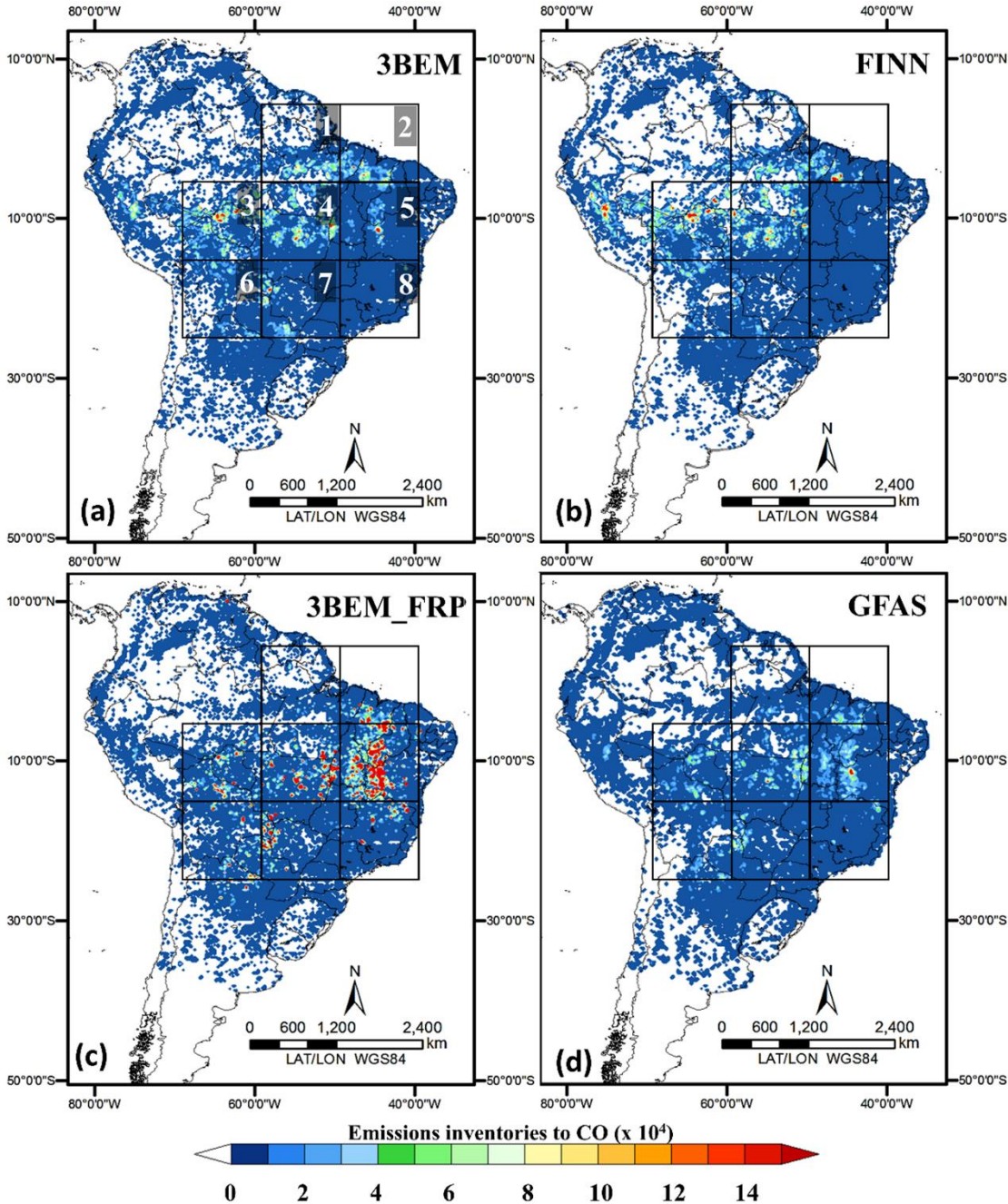

Figure 3. Spatial distribution of total CO emissions ($10^4$ kg.m$^{-2}$) over South America from 1st September to 31st of October 2012, estimated by (a) 3BEM, (b) FINN, (c) 3BEM_FRP, and (d) GFAS.

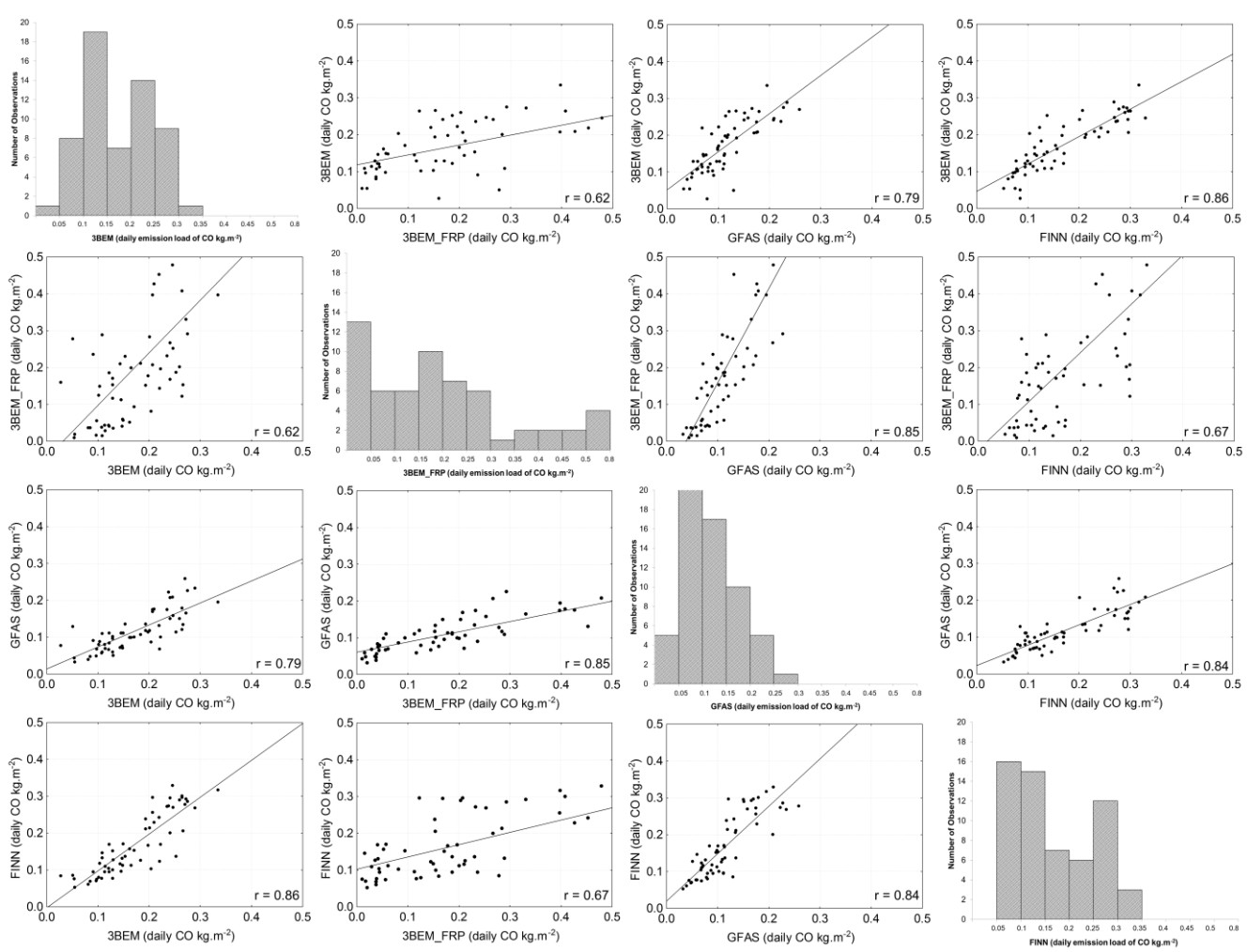

Figure 4. Linear regression between daily areal-averaged emission load of CO to South America from 3BEM, 3BEM_FRP, FINN and GFAS examined between 1st September to 31st of October 2012. In the graph, x and y-axis represent the CO (kg.m$^{-2}$) of each fire inventory (i.e. the first line of 3BEM regressions, 3BEM_FRP, GFAS and FINN are x-axis).

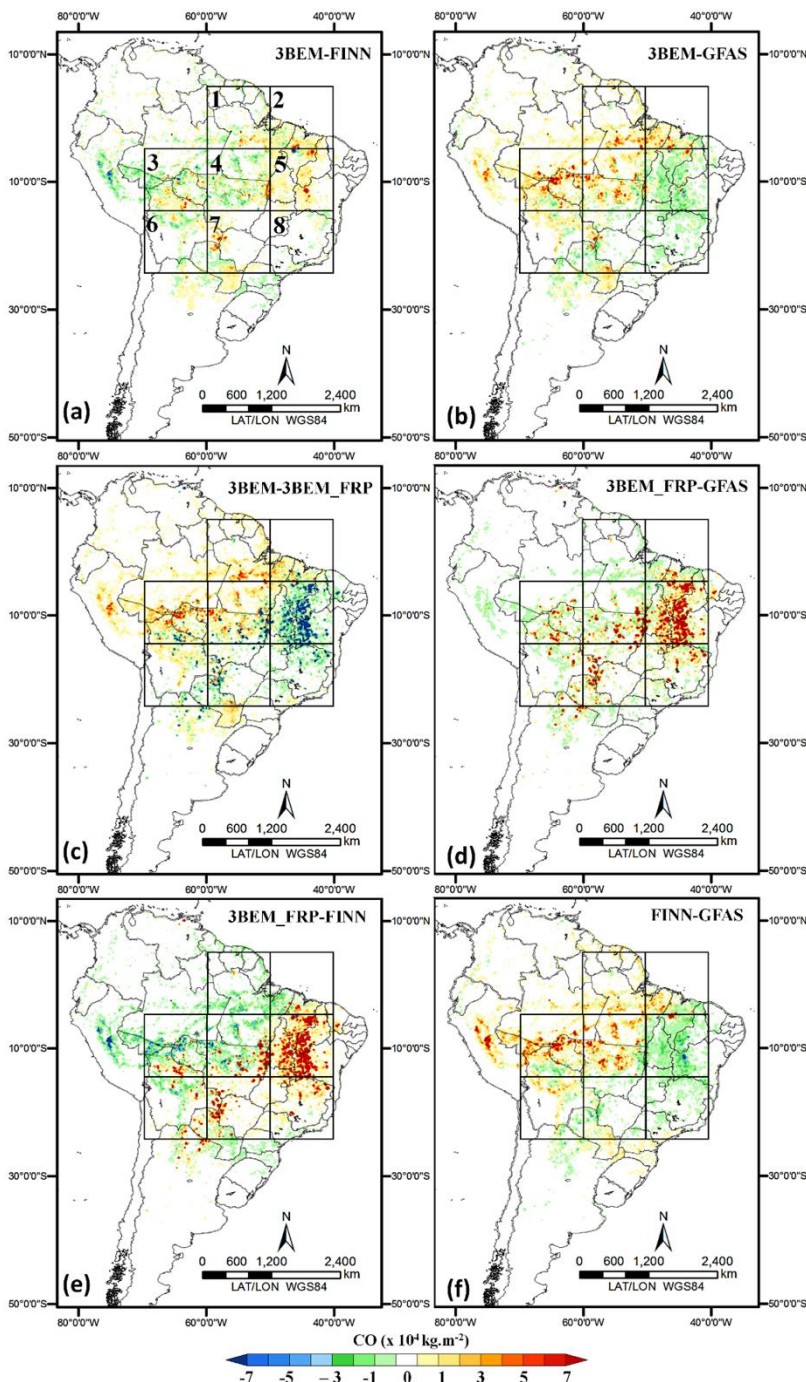

Figure 5. Difference between fire inventories. A) 3BEM-FINN; B) 3BEM-GFAS; C) 3BEM-3BEM_FRP; d) 3BEM_FRP-GFAS; e) 3BEM_FRP-FINN; f) FINN-GFAS.

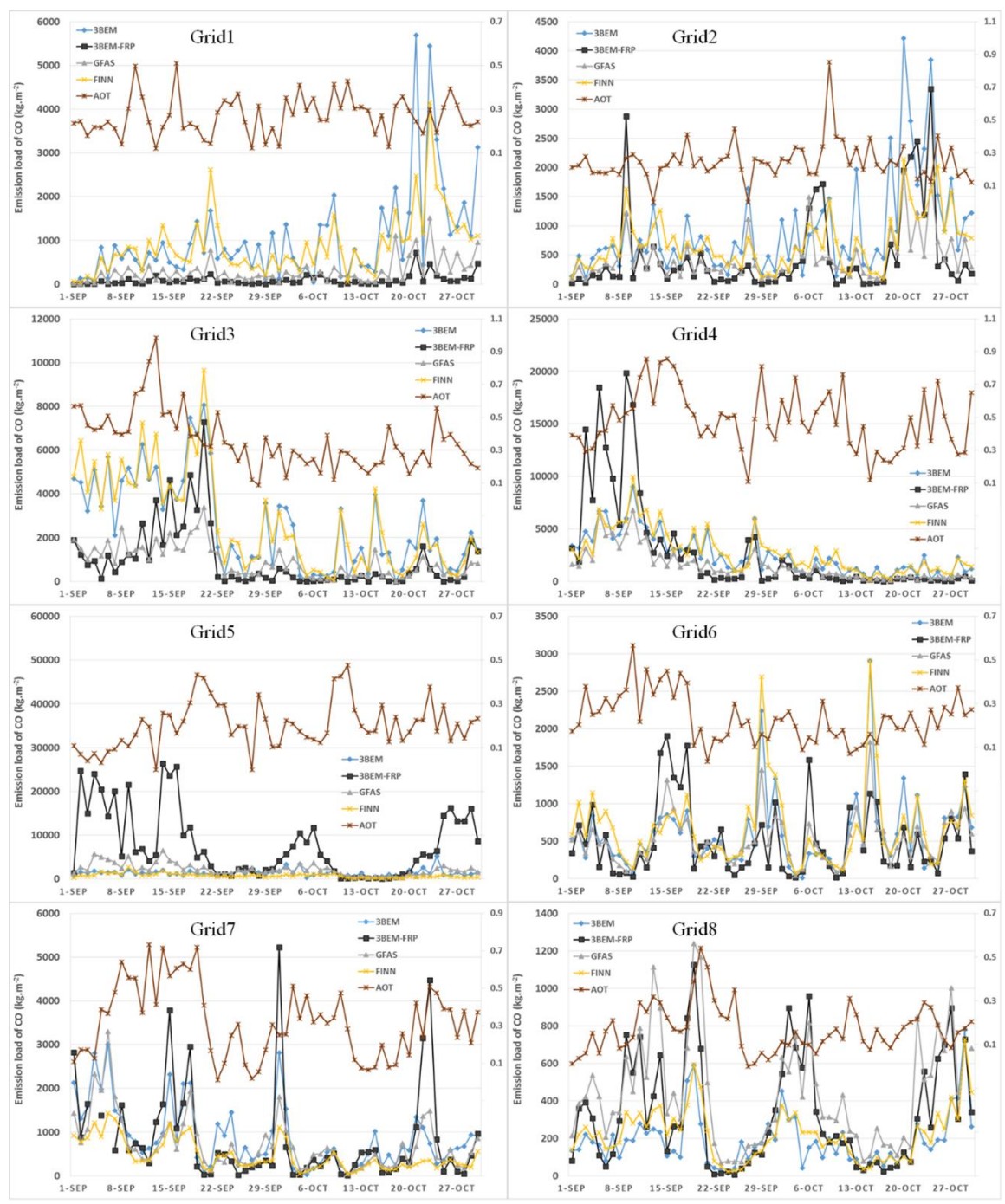

Figure 6. Lat-Lon Average time series of Aqua MODIS AOT$_{550nm}$ and 3BEM, 3BEM_FRP , GFAS, and FINN emission load of CO time series.

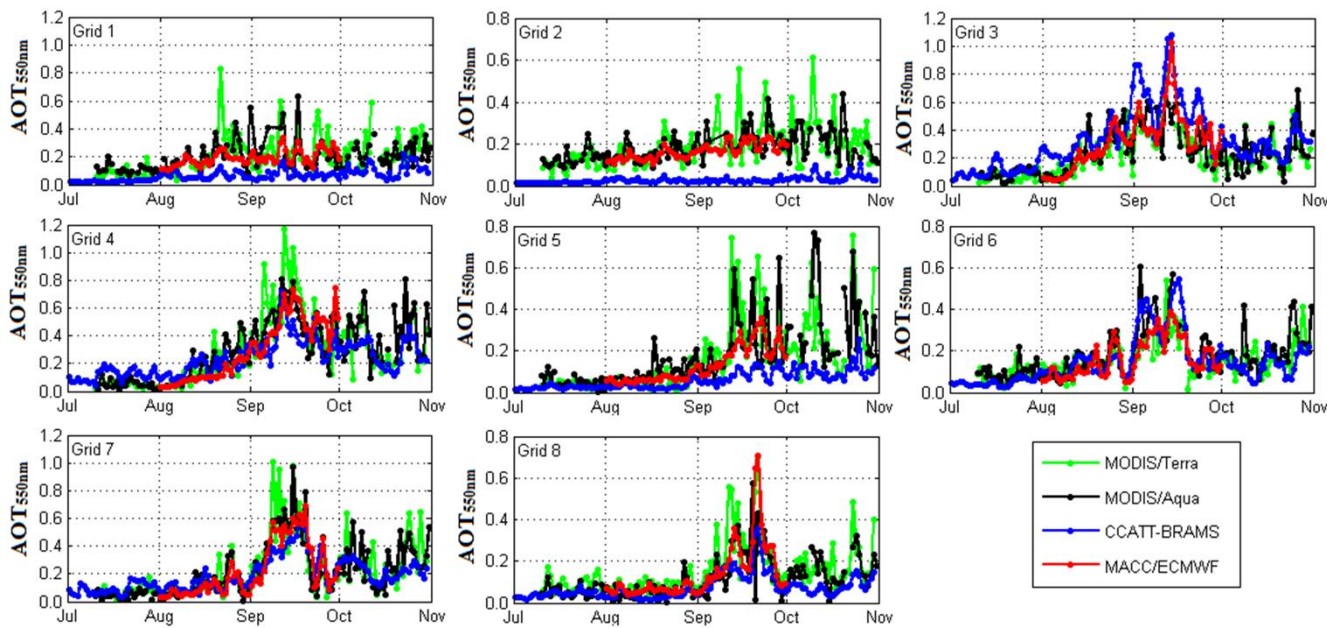

5    Figure 7. Lat-Lon Average time series of Terra/MODIS $AOT_{550nm}$ (in green), Aqua/MODIS $AOT_{550nm}$ (in black), CCATT-

BRAMS simulated $AOT_{550nm}$ (in blue) and MACC/ECMWF (in red) $AOT_{550nm}$ analysis.  Grids are those show in Figure 2.