# Peer review of "Assessment of fire emissions inventories during the South American Biomass Burning Analysis (SAMBBA) experiment"

_Atmospheric Chemistry and Physics, 2015_

## Referee Comment (RC1) · Anonymous Referee #1 · 11 Mar 2016

This manuscript reviews several methods of estimating emissions from fires in Brazil originating from land use activities. The presentation does a very good job reviewing the various methods and products used in regional and global fire emissions inventories.

The paper does a good job of concisely presenting the results of this analysis. The figures are generally clear (see note on Fig 3, however), and the paper has value to the community. Two specific points:

Figure 3 – The order of presentation does not match the previous figure or the text. Please re-order these plots. Also, the plot are lacking axis labels. I understand this allows the efficient display, but it takes some time to understand this. The caption

[Figure]

helps the reader understand the layout and why there are not labels, but needs a bit more explanation, since there are no values associated with the data points.

I would suggest that the results & discussion section be broken into sub-headings to help the reader. The final two paragraphs are more specifically discussion, as it presents more than the emissions results, and explains why these data are important to estimate accurately.

---

## Referee Comment (RC2) · Anonymous Referee #2 · 25 Mar 2016

The study by Pereira compares CO emissions from four different biomass burning emission inventories (spatially and temporally) over South America during the SAMBBA field campaign. The study also makes an attempt to evaluate these emissions in two chemistry transport models against MODIS AOT. The concept of the study is well within the scope of ACP and will be very useful to users of these emissions, particularly atmospheric chemistry and aerosol modellers. The results of the study are generally well described and summarised. However, I think there can be some improvements made in the layout of the manuscript (text, figures and tables) and the clarity of the text. I recommend publication in ACP once the following general and specific comments have been addressed.

[Figure]

General comments

1. To compare the emission inventories, the authors rely on the use of correlation between the different datasets. Correlation is a useful metric, but the authors need to be clearer in the manuscript what type of correlation they have calculated or between what data points i.e. state whether the correlation is temporal or spatial. This needs to be improved both in the text (abstract, Sect. 3, conclusions. . .) and in the captions for Table 2 and Figure 4. Also, it would be useful if the correlation values were accompanied with an estimate of bias (absolute or percentage etc.) so that the reader can judge how the magnitudes of the emissions compare quantitatively.

2. I believe the most important and useful part of the manuscript is the evaluation of the emissions against observations. This section has the potential to inform the reader of the most appropriate emission inventory to use for this region/time period. However, this section falls slightly short. Firstly, the description of the models and the emissions they use is too brief. What is the spatial resolution of the models? Are the fire emissions scaled at all? What version of GFAS is used in MACC? Are the emissions injected all at the surface in the models? Secondly, no statistics are computed to compare the model and observations. It would be useful to have correlation and/or bias values calculated to quantify the agreement (as for the comparison of emission inventories). It would also be useful to see the difference between model and observations mapped as in Fig 5. Finally, what is the reason for the underestimation of AOT in Grid 5? Would 3BEM-FRP emissions perform better here?

3. The results and discussion section should be broken down into sub-sections to improve readability of the text. I suggest separate sections on: i) comparing emission inventories spatially; ii) comparing emissions temporally; iii) evaluating the emissions against observations. The figures and tables should be re-arranged accordingly. Also, make sure it is clear which values in the text are shown in which figure/table, this is not always the case (see specific comments below).

4. There are several typos, grammar mistakes, naming inconsistencies and missing words in the text that at times distracted me from the science content of the paper. There were too many errors to point out individually in the specific comments below so the authors should thoroughly proofread the revised manuscript and make the necessary corrections prior to re-submission.

Specific comments

1. Abstract, L26-33: Please state clearly what the under/overestimations of the emissions are relative to i.e. to the other inventories NOT observations. Also, there is no mention of the results of the comparison to observed AOT.

2. Introduction (or Sect. 2.1): The positives, negatives and key uncertainties in the biomass burning emission estimation methods should be discussed and compared either in the introduction or later in Sect. 2.1 (some points for burnt area given i.e. not near real time, but more could be added). Also, I'm also surprised that GFED emissions are not mentioned in the text, since these emissions are widely used in models.

3. Sect. 2.1.1: How is cloud cover dealt with in the 3BEM and 3BEM-FRP inventories? Are any corrections made for missed detections due to cloud (or smoke) cover? Apologies if I missed this information.

4. Sect. 3, P7, L19-24: Can you suggest any reasons for the large differences between emissions in grid 5? In particular, why are FINN emissions so low relative to the other emissions? There must be active fire detections here for the FRP-based inventories to predict emissions? Could the assumed fire size be too small in FINN? (Although this seems unlikely if fires in this region are mainly agricultural).

5. Sect. 3, P7, L25. . .: Is this the correlation between the spatial patterns of the emissions (see general comment 1)? Please clarify in the text.

6. Sect. 3, P8, L1. . .: It's confusing to start referring to the emission inventories as models here. I suggest sticking to "emission inventory" so as not to confuse it with

the term "models" used when discussing chemistry transport models in the final few paragraphs of this section.

7. Sect. 3, P8, L1 onwards: (As for general comment 1) what are these correlations calculated between? Is it the spatial correlation being discussed? What figure table do these correlation values link to?

8. Sect 3., P8, L2: From Table 3 it looks like total CO emissions from FINN are lower than 3BEM in 4 out of 8 grids, so this statement doesn't seem to follow. Which figure/table are you referring to here? Please clarify. (Cannot tell from Fig 4 because there are no values on the axes).

9. Sect. 3, P8, L11: Table 3 shows total CO emissions from GFAS are lower than 3BEM in grid 2. Do you mean grid 8?

10. Sect., P9, L18-19: How is this agreement "good", please quantify or explain. Some peaks in CO emission and AOT look anti-correlated on a few of the days. In what way would you expect CO emissions to agree with retrieved AOT?

11. Conclusions, P11, L4: "...where 3BEM_FRP and GFAS deliver a better accuracy". How do you show that 3BEM_FRP and GFAS emissions deliver a better accuracy? Is this surmised from the comparison between MACC+GFAS and MODIS AOT?

12. Conclusions: Perhaps I have misunderstood something but the final two sentences do not seem to follow on from the previous sentence.

13. Figure 4: This figure is difficult to interpret and needs a clearer/more detailed caption. What does each data point represent? What do the grey bars represent? Axes labels and values would aid interpretation.

---

## Author Comment (AC1) · 5 May 2016

Dear Reviewer, First, we would like to thank you for the helpful comments and suggestions. Below, we describe the changes made and the answers (point-by-point).

Referee # 1

RC1: "Figure 3 – The order of presentation does not match the previous figure or the text. Please re-order these plots. Also, the plot are lacking axis labels. I understand this allows the efficient display, but it takes some time to understand this. The caption helps the reader understand the layout and why there are not labels, but needs a bit more explanation, since there are no values associated with the data points."

[Figure]

**ACPD**

AC1: We have changed the figure layout. To a better understanding we added axis labels to all graphs of the figure including the data points values.

RC2: "I would suggest that the results & discussion section be broken into sub-headings to help the reader. The final two paragraphs are more specifically discussion, as it presents more than the emissions results, and explains why these data are important to estimate accurately."

AC2: We divided the topic results & discussion section into three sub-topics: 3.1 Comparing emission inventories spatially; 3.2 Assessment of fire inventories with AOT; 3.3 Evaluating the emissions against observations.
* * *

---

## Author Comment (AC2) · 5 May 2016

Dear Reviewer,

First, we would like to thank you for the helpful comments and suggestions. Below, we describe the changes made and the answers (point-by-point).

Referee # 2

General comments RC1: "To compare the emission inventories, the authors rely on the use of correlation between the different datasets. Correlation is a useful metric, but the authors need to be clearer in the manuscript what type of correlation they have calculated or between what data points i.e. state whether the correlation is temporal or

spatial. This needs to be improved both in the text (abstract, Sect. 3, conclusions. . .) and in the captions for Table 2 and Figure 4. Also, it would be useful if the correlation values were accompanied with an estimate of bias (absolute or percentage etc.) so that the reader can judge how the magnitudes of the emissions compare quantitatively."

AC1: The article was revised. We changed all correlations to Linear correlation. Also, we described the type of data used to perform linear correlation (daily areal-averaged) and calculate the absolute bias for the four inventories to the eight grids.

RC2: "I believe the most important and useful part of the manuscript is the evaluation of the emissions against observations. This section has the potential to inform the reader of the most appropriate emission inventory to use for this region/time period. However, this section falls slightly short. Firstly, the description of the models and the emissions they use is too brief. What is the spatial resolution of the models? Are the fire emissions scaled at all? What version of GFAS is used in MACC? Are the emissions injected all at the surface in the models? Secondly, no statistics are computed to compare the model and observations. It would be useful to have correlation and/or bias values calculated to quantify the agreement (as for the comparison of emission inventories). It would also be useful to see the difference between model and observations mapped as in Fig 5. Finally, what is the reason for the underestimation of AOT in Grid 5? Would 3BEM-FRP emissions perform better here?

AC2: The models configurations were further described in section 2.3. There is another complementary paper (to be submit) analyzing the differences between model and observation AOD fields and trying to the evaluate the potential role of emission and meteorology combination on models performance (compared with other variables). MACC model uses AOD assimilation based on MODIS products in order to improve the representation of model AOD field. Therefore, in this case, it is more difficult to assess either the role of meteorology or emissions inventories on its performance. . CCATT-BRAMS does not use assimilation, which allows, to some degree, an evaluation of the influence of meteorological and emission aspects. Furthermore, for the SAMBBA experiment, it was not possible to run a designed experiment specifically to evaluate the solely impacts of the four emissions inventories on the SAMBBA chemistry transport models performance due to computational limitations. Nevertheless, current results highlight and quantify divergences among emission inventories currently used world-wide to simulate smoke emission. The presented comparison between AOT observed and simulated by the chemistry transport models(CTM) it is not the ideal to evaluate the emission inventories, but, at least for CCATT-BRAMS, it provide a perspective of the CTM current performance on simulating South America smoke spatial and temporal variability given the emission inventory used to feed the model.

RC3: "The results and discussion section should be broken down into sub-sections to improve readability of the text. I suggest separate sections on: i) comparing emission inventories spatially; ii) comparing emissions temporally; iii) evaluating the emissions against observations. The figures and tables should be re-arranged accordingly. Also, make sure it is clear which values in the text are shown in which figure/table, this is not always the case (see specific comments below)."

AC3: We broke the results & discussion section into three new topics: 3.1 Comparing emission inventories spatially; 3.2 Assessment of fire inventories with AOT; 3.3 Evaluating the emissions against observations.

RC4: "There are several typos, grammar mistakes, naming inconsistencies and miss-ing words in the text that at times distracted me from the science content of the paper. There were too many errors to point out individually in the specific comments below so the authors should thoroughly proofread the revised manuscript and make the neces-sary corrections prior to re-submission."

AC4: All suggestions were accepted and a proofreading performed.

Specific comments

1. Abstract, L26-33: Please state clearly what the under/overestimations of the emissions are relative to i.e. to the other inventories NOT observations. Also, there is no mention of the results of the comparison to observed AOT.

ASC1: The abstract was modified and the results of the comparison to observed AOT are now described. Also, we have adapted the abstract to clarify the text.

2. Introduction (or Sect. 2.1): The positives, negatives and key uncertainties in the biomass burning emission estimation methods should be discussed and compared either in the introduction or later in Sect. 2.1 (some points for burnt area given i.e. not near real time, but more could be added). Also, I'm also surprised that GFED emissions are not mentioned in the text, since these emissions are widely used in models.

ASC2: We inserted two paragraphs in Introduction about the uncertainties. Also, we mentioned the GFED in introduction.

3. Sect. 2.1.1: How is cloud cover dealt with in the 3BEM and 3BEM-FRP inventories? Are any corrections made for missed detections due to cloud (or smoke) cover? Apologies if I missed this information.

ASC3: In 3BEM or 3BEM_FRP, when the fire location is undetected due to cloud cover this fire was not accounted in biomass burning estimation. However, due to the frequency of observations (mainly in GOES and SEVIRI) 3BEM needs only one fire detection in the lifecycle of the event due to filtering process. Also, in 3BEM_FRP, if the active fire has no observations in four hours we assume that the fire is over. Thus, missed detection due cloud have impact in FRE integration if the cloud persists for more than 8 satellite acquisitions. Also, a 10% cloud error could be associated to this phenomenon. We inserted this explanation in the text.

4. Sect. 3, P7, L19-24: Can you suggest any reasons for the large differences between emissions in grid 5? In particular, why are FINN emissions so low relative to the other emissions? There must be active fire detections here for the FRP-based inventories to predict emissions? Could the assumed fire size be too small in FINN? (Although this
seems unlikely if fires in this region are mainly agricultural).

ASC4: Probably the main cause of low emissions of FINN is associated with MODIS fire detections in this area. The fire cycle starts frequently in the middle of the afternoon (after 15:00 local time). In some cases, the fire end before the next MODIS acquisition, missing a hotspot. When one uses GOES and AVHRR data, such as does 3BEM, we increase the fire emission estimation, however, in this area (in fire season) we have a low cloud cover, that could improve the estimation due temporal integration of FRP methods.

5. Sect. 3, P7, L25: : :: Is this the correlation between the spatial patterns of the emissions (see general comment 1)? Please clarify in the text.

ASC5: We have changed all correlations to Linear correlation and inserted the type of data used, such as daily areal-averaged.

6. Sect. 3, P8, L1: : :: It's confusing to start referring to the emission inventories as models here. I suggest sticking to "emission inventory" so as not to confuse it with the term "models" used when discussing chemistry transport models in the final few paragraphs of this section.

ASC6: We have changed all to emission inventories.

7. Sect. 3, P8, L1 onwards: (As for general comment 1) what are these correlations calculated between? Is it the spatial correlation being discussed? What figure table do these correlation values link to?

ASC7: We have changed all correlations to Linear correlation and inserted the type of data used, such as daily areal-averaged. The mean correlation are described in table 2.

8. Sect 3., P8, L2: From Table 3 it looks like total CO emissions from FINN are lower than 3BEM in 4 out of 8 grids, so this statement doesn't seem to follow. Which figure/ table are you referring to here? Please clarify. (Cannot tell from Fig 4 because there

are no values on the axes).

ASC8: We inserted the values and changed the figure 4. In most grids FINN present lower values, but when the bootstrap is performed, the error analysis indicates that in the other grids the difference is more accentuated indicating an overall trend of overestimation.

9. Sect. 3, P8, L11: Table 3 shows total CO emissions from GFAS are lower than 3BEM in grid 2. Do you mean grid 8?

ASC9: We corrected the information.

10. Sect., P9, L18-19: How is this agreement "good", please quantify or explain. Some peaks in CO emission and AOT look anti-correlated on a few of the days. In what way would you expect CO emissions to agree with retrieved AOT?

ASC10: We changed the sentence to: "in the general pattern of temporal evolution we could observe a good agreement between CO emission load estimated by the four emission inventories and MODIS AOT550nm"

11. Conclusions, P11, L4: ": where 3BEM_FRP and GFAS deliver a better accuracy". How do you show that 3BEM_FRP and GFAS emissions deliver a better accuracy? Is this surmised from the comparison between MACC+GFAS and MODIS AOT?

ASC11: Yes, there is not a clear basis to conclude this, therefore we excluded this sentence.

12. Conclusions: Perhaps I have misunderstood something but the final two sentences do not seem to follow on from the previous sentence.

ASC12: The two sentences were inserted in the sequence of the above paragraph.

13. Figure 4: This figure is difficult to interpret and needs a clearer/more detailed caption. What does each data point represent? What do the grey bars represent? Axes labels and values would aid interpretation.
Interactive comment

ASC13: We inserted the values and changed the figure 4. Also, we inserted the article with track changes.

Please also note the supplement to this comment:
http://www.atmos-chem-phys-discuss.net/acp-2015-1053/acp-2015-1053-AC2-supplement.pdf

[Figure]

**Supplement:**

[revised manuscript text omitted]